# Validation testing to determine the sensitivity of lateral flow testing for asymptomatic SARS-CoV-2 detection in low prevalence settings: Testing frequency and public health messaging is key

Jack Ferguson[1☯], Steven Dunn[2☯], Angus Best[3☯], Jeremy Mirza[3], Benita Percival[3], Megan Mayhew[3], Oliver Megram[3], Fiona Ashford[3], Thomas White[3], Emma Moles-Garcia[3], Liam Crawford[3], Tim Plant[3], Andrew Bosworth[4], Michael Kidd[5], Alex Richter[1], Jonathan Deeks[6,7], Alan McNally[2]*

1 Institute of Cancer and Genomic Science, College of Medical and Dental Science, University of Birmingham, Birmingham, United Kingdom, 2 Institute of Microbiology and Infection, College of Medical and Dental Science, University of Birmingham, Birmingham, United Kingdom, 3 Clinical Immunology Service, Institute of Immunology and Immunotherapy, College of Medical and Dental Science, University of Birmingham, Birmingham, United Kingdom, 4 University Hospital Birmingham NHS Foundation Trust, Birmingham, United Kingdom, 5 Public Health England, Birmingham, United Kingdom, 6 Institute of Applied Health Research, College of Medical and Dental Science, University of Birmingham, Birmingham, United Kingdom, 7 NIHR Birmingham Biomedical Research Centre, University Hospitals Birmingham NHS Foundation Trust and University of Birmingham, United Kingdom

☯ These authors contributed equally to this work.
* a.mcnally.1@bham.ac.uk

## Abstract

Lateral flow devices (LFDs) are quickly being implemented for use in large-scale population surveillance programs for SARS-CoV-2 infection in the United Kingdom. These programs have been piloted in city-wide screening in the city of Liverpool and are now being rolled out to support care home visits and the return home of University students for the Christmas break. Here, we present data on the performance of LFDs to test almost 8,000 students at the University of Birmingham between December 2 and December 9, 2020. The performance is validated against almost 800 samples using PCR performed in the University Pillar 2 testing lab and theoretically validated on thousands of Pillar 2 PCR testing results performed on low-prevalence care home testing samples. Our data show that LFDs do not detect infections presenting with PCR Ct values over 29 to 30 as determined using the Thermo Fisher TaqPath asssay. This may be of particular importance in detecting individuals that are either at the early, or late stages of infection, and reinforces the need for frequent, recurrent testing.

## Introduction

In November 2020, the United Kingdom government announced a plan to introduce mass-scale population testing for Severe Acute Respiratory Syndrome Coronavirus 2 (SARS-CoV-2)

**Data Availability Statement:** All relevant data are within the paper and its Supporting Information files.

**Funding:** The PCR testing in this manuscript is funded by the UK Department for Health and Social Care (DHSC) as part of pillar 2 testing, in an award made directly to the University of Birmingham. The provision of LFD tests is funded by DHSC as part of a national student testing program, and funded directly to the University of Birmingham. DHSC have had no role in in study design, data collection and analysis, decision to publish, or preparation of the manuscript.

**Competing interests:** The authors have declared that no competing interests exist.

**Abbreviations:** COG-UK, COVID-19 Genomics UK Consortium; DHSC, Department for Health and Social Care; LFD, Lateral Flow Device; LoD, Limit of Detection; RT-PCR, reverse transcription PCR; SARS-CoV-2, Severe Acute Respiratory Syndrome Coronavirus 2; VOCs, variants of concern.

infection using Lateral Flow Devices (LFDs) [1]. The most commonly utilised of these is an LFD manufactured and marketed by Innova Medical Group, a subsidiary of Xiamen Biotime Biotechnology Company, which was the first device to pass a 4-phase validation. The LFD is a rapid lateral flow device based on colloidal gold immunochromatography designed to detect the presence of SARS-CoV-2 nucleocapsid antigens in nasopharyngeal swabs [2]. The test can provide a result within 30 minutes allowing rapid testing on a mass scale.

The Innova LFD has very quickly been put into implementation by the Department for Health and Social Care (DHSC) and was employed in the city of Liverpool to deliver an ambitious mass-scale surveillance project of the city over a 2-week period [3]. Data from the city council [4] show that 71,684 LFD tests were performed alongside 51,855 gold-standard PCR tests (a total of 119,054 residents tested) with 439 people testing positive (0.37% positivity rate). The LFD tests are now being used in to support people visiting relatives in Care Homes and are being rolled out to support testing of University students and secondary school pupils across the UK [1].

In order to support the use of the Innova LFD test, the University of Oxford and Public Health England (PHE) performed a series of validation trials of the LFD, benchmarking their performance against reverse transcription PCR (RT-PCR) using swabs from a number of research trials in the UK [5]. These included comparative testing on samples taken for the NIHR FALCON study [6], which aims to evaluate diagnostic platforms [6], and bespoke trials including PHE, hospital, military staff, and schools. The key headline findings of the validation report were that the LFD had a Limit of Detection (LoD) of around 100 plaque forming units/ ml or 100,000 RNA copies/ml [7]. In the report, it is not made clear which RT-PCR assay is used in the comparison, but the cycle threshold (Ct) value given of 25.5 equating to 100 pfu/ml is not obtained from the ThermoFisher COVID-19 TaqPath assay [8] employed in the majority of Pillar 2 testing labs in the UK. As such, the validation report may not fully indicate the potential of the performance of the LFD against the vast majority of COVID PCR testing done in the UK through Pillar 2. This study aimed to assess the efficacy of the Innova LFD platform for the purposes of mass screening of asymptomatic individuals, using a purpose-built facility at the University of Birmingham. We determine through RT-PCR validation that Innova LFD platform captures all cases in which individuals have high quantities of viral RNA present on the swab, as presented to us (i.e., N gene Ct <30) and recommend that frequent, routine testing, paired with clear public health messaging, is key to the successful implementation of LFD testing in order to reduce overall community burden of SARS-CoV-2.

## Methods

### Lateral flow device testing of students

As part of the national plan to test students for SARS-CoV-2 before the 2020 Christmas break, the DHSC provided University of Birmingham with Innova LFDs to test 15,000 students. Using a University-owned booking system, a total of 7,189 students were tested between December 2 and December 9. Students were provided with a sterile nasopharyngeal swab and under supervision from a trained member of the University testing team, swabbed both tonsils and a single nasal cavity. The swab was passed through an opening in a plastic screen to a designated test area, where it was immediately processed according to the Innova protocol [2]. Briefly, this involves adding 6 drops of sample buffer to a sample tube, immersing and agitating the patient's swab inside the buffer to homogeneity, and loading 2 drops of the final solution onto the test cartridge. Tests were performed by trained members of the University testing team drawn from postgraduate and final year undergraduate students in the College of Medical and Dental Science in the University, supervised by highly experienced postdoctoral

researchers. In total, a team of 18 test operatives oversaw 36 testing booths, with a student attending a booth every 10 minutes. A further 7 staff acted as results recorders logging the test results via a barcode through a mobile phone device and result recording app, which were both supplied by DHSC.

## Validation of lateral flow device test results by PCR testing

University of Birmingham is home to a national Pillar 2 testing laboratory (i.e., swab testing for the wider population), termed Turnkey lab, which conducts SARS-CoV-2 PCR diagnostics on behalf of DHSC [9]. The laboratory uses the ThermoFisher COVID-19 TaqPath assay used routinely in the Lighthouse laboratory testing network and tests a range of samples from mobile and stationary test sites [9]. On each day of testing, 90 residual LFD test samples (buffer solution in which the nasopharyngeal swab is resuspended to perform the test) were selected for confirmatory PCR testing, which constituted a single 96-well plate including positive and negative controls. All positive samples were chosen for confirmatory PCR, and the remainder were randomly selected samples. All samples were completely anonymous to the testing team with no identifying labels and were arbitrarily numbered from 1 to 90 each day. Sterile water (approximately 350 μl) was added to the samples to bring them to the 500 microlitres required for automated RNA extraction and tested according to Pillar 2 laboratory protocol [9]. The use of anonymised waste samples from student testing in this study was allowed under ethics gained to aid assay development (NRES Committee West Midlands—South Birmingham 2002/201 Amendment Number 4, 24 April 2013).

## Statistical analysis of PCR validation

The efficient stratified study design involved verification of all Innova test positives with RT-PCR and a random sample of 720/7,187 Innova test negatives. Estimates of sensitivity and prevalence with 95% confidence intervals were obtained using maximum likelihood inverse probability weighted logistic regression to account for the sampling design, with conversion of the estimated odds to probabilities. Weights of 1 for Innova test positives and 9.98 (7,187/720) for Innova test negatives were used. Expected numbers of cases were computed from the prevalence estimate. The estimate of specificity was obtained without weighting as no Innova test positives were observed in those with negative PCR. Exact binomial methods were used to compute confidence intervals for test yield and specificity.

## Theoretical validation of LFD performance against Pillar 2 PCR test data

As part of Pillar 2 testing, our Turnkey laboratory also conducts PCR testing as part of the national Care Home surveillance plan implemented by DHSC to test all care home staff and residents to assist in control of COVID-19 transmission in UK care homes [10]. Between October 25 and November 5, the Birmingham Turnkey laboratory processed a total of 19,176 PCR tests on home test and care home samples from across the UK. Of these, 641 samples tested positive for SARS-CoV-2 using the cutoff of 2 of 3 gene targets amplifying at a Ct value of 35 or under [9]. This gives a positivity rate of 3.3%, around the rate that might be reasonably be expected in a large random surveillance of the UK population at that moment in time.

## Results

### Lateral flow testing of University of Birmingham students

A total of 7,189 students voluntarily attended the asymptomatic student testing centre between December 2 and December 9. Students were refused a test if they had any symptoms of

**Table 1. Table of results for Lateral Flow Device testing of University of Birmingham students and confirmatory PCR testing of approximately 10% of samples.**

| Day | LFD tests | | | PCR | Validation |
|---|---|---|---|---|---|
| | **Positive** | | **Negative** | **Positive** | **Negative** |
| 02/12/2020 | 0 | | 630 | 0 | 90 |
| 03/12/2020 | 2 | | 589 | 2 | 89 |
| 04/12/2020 | 0 | | 1,102 | 1 | 88 |
| 05/12/2020 | 0 | | 860 | 1 | 89 |
| 06/12/2020 | 0 | | 610 | 0 | 90 |
| 07/12/2020 | 0 | | 813 | 2 | 88 |
| 08/12/2020 | 0 | | 1,259 | 2 | 88 |
| 09/12/2020 | 0 | | 1,320 | 0 | 90 |
| **Totals** | **2** | | **7,183** | **8** | **712** |

COVID-19 and were referred to a local test site for PCR testing. Results of 4 samples were void (as defined by the manufacturer's protocol [2]), and 2 samples tested positive for SARS-CoV-2 by lateral flow, a prevalence of 0.03% (95% CI 0.02% to 0.10%) in the students volunteering for a test (Table 1).

## Lateral flow results validation by pillar 2 PCR

The 2 samples positive by Lateral Flow and another 718 randomly selected negative samples were transported to the University Turnkey laboratory for PCR testing (9.9% of sample total). Of the 720 samples tested by PCR, 8 were positive: The 2 positive by Lateral Flow and a further 6 samples negative on the LFD (Table 1).

Our PCR validation data suggest a true prevalence rate in the student population tested of 0.86% (95% CI 0.40% to 1.86%). The overall sensitivity of the test in the tested student population was observed to be 3.23% (95% CI 0.60% to 15.59%). We estimate that there would have been 62 cases in the 7,185 students, of which 60 were missed. There were no false positive results, observed specificity was 100% (95% CI 99.48% to 100.00%).

We further investigated the PCR testing discrepancy by extracting the Ct values for the amplification curves for the 8 PCR positive samples (Table 2). Our data show that the 6 samples testing false negative by Lateral Flow all had Ct values >29, while the 2 true positive samples had Ct values of 20 and 25. We collated the RT-PCR raw data from 5 technical replicates of assays performed on the Qnostics SARS-CoV-2 analytical Q-panel– 01 [11] and generated average Ct values for each of the known viral titres provided in the panel (Fig 1). Using these data, we determined that, at 100 viral copies per ml (the designated LoD for the Innova LFD

**Table 2. Pillar 2 PCR Ct values of confirmatory samples positive for SARS-CoV-2.** The samples which tested positive on Lateral Flow device are in grey columns.

| Well Number | ORF1ab Ct | N gene Ct | S gene Ct |
|---|---|---|---|
| 38 | 32.501118 | 33.89259 | 34.815563 |
| 1 | 25.160471 | 25.678833 | 25.034386 |
| 57 | 21.319279 | 22.638311 | 20.582413 |
| 36 | 28.538937 | 29.359957 | 29.123411 |
| 38 | | 32.216614 | 33.92468 |
| 42 | 27.669174 | 29.401642 | 27.895124 |
| 14 | 30.770878 | 31.718863 | 31.998856 |
| 34 | 30.917858 | 31.794565 | 31.35588 |

| QNostic Sample ID | Copies/ml | Log10 Copies | Orf1ab | S Gene | N Gene |
|---|---|---|---|---|---|
| SCV2AQP01-S01 | 1000000 | 6 | 15.3 (100%) | 16.5 (100%) | 15.8 (100%) |
| SCV2AQP01-S02 | 100000 | 5 | 18.3 (100%) | 20.5 (100%) | 17.4 (100%) |
| SCV2AQP01-S03 | 10000 | 4 | 22.8 (100%) | 24.9 (100%) | 23.6 (100%) |
| SCV2AQP01-S04 | 5000 | 3.7 | 19.5 (100%) | 24.8 (100%) | 24.1 (100%) |
| SCV2AQP01-S05 | 1000 | 3 | 25.9 (100%) | 28.6 (100%) | 26.6 (100%) |
| SCV2AQP01-S06 | 500 | 2.7 | 25.8 (100%) | 29.1 (100%) | 25.8 (100%) |
| SCV2AQP01-S07 | 100 | 2 | 27.7 (33%) | 30.3 (66%) | 30.8 (100%) |
| SCV2AQP01-S08 | 50 | 1.7 | 29.3 (17%) | 31.1 (27%) | 29.1 (55%) |
| SCV2AQP01-S09 | Negative | ---- | Negative | Negative | Negative |

**Fig 1. Analytical sensitivity and specificity of the Birmingham Turnkey lab RT-PCR pipeline.** This was assessed against the commercial Qnostics SARS-CoV-2 analytical Q-panel– 01. Ct values are a median of 5 independent technical replicates, and figures in parentheses indicate the percentage of replicates returning a PCR positive for that given gene target (Ct <35). Ct, cycle threshold; RT-PCR, reverse transcription PCR; SARS-CoV-2, Severe Acute Respiratory Syndrome Coronavirus 2.

[2,7]), the equivalent Ct values for the Pillar 2 PCR assay would be a Ct of 30.8 based on the N gene target, roughly in line with our PCR validation data.

## Sensitivity of lateral flow device in the student population by Ct value

From our data, the LFD test yield is 2.8 per 10,000 (0.3 to 10 per 10,000 tests). The sensitivity of the LFD in the tested population differs greatly dependent on the viral titre of the person tested. At a PCR Ct value <29, the sensitivity was 100% (95% CI 15.8 to 100). However, at a Ct < = 29, this dropped to 9.1% (1.03 to 49.1), and at Ct <33 dropped again to 5.01% (0.78 to 32.14).

## Extrapolation of pillar 2 PCR data to theoretically evaluate lateral flow device performance

We collated the raw RT-PCR data for all 641 of our positive samples as of November 5 and ranked them according the N gene Ct value (S1 Table). We then plotted the distribution of Ct values for our 641 positive samples (Fig 2). Using the LoD of 100 pfu/ml, we determined that this would correlate with an N gene Ct value of 30.8 plus one other gene target amplifying at a Ct <35. By applying this theoretical level of performance to the LFD, we determine that 99 of our positive samples would not be able to be detected by the Innova LFD given that the Ct value of N gene is above 30.8. This equates to 15.44% of our true positive RT-PCR samples would not be detected using the Innova LFD. This means that, theoretically, the Innova LFD (when compared to Pillar 2 samples from low-prevalence, asymptomatic population screening similar to student and care home surveillance) would successfully detect 84.56% of all infections.

## Discussion

Expansion of mass testing for SARS-CoV-2 is an issue many countries are facing, with LFDs seen as a viable accompaniment to the gold standard of RT-PCR tests as a way to increase capacity and screen asymptomatic populations. The University of Birmingham deployed a purpose built testing facility in order to screen its student population before they travelled

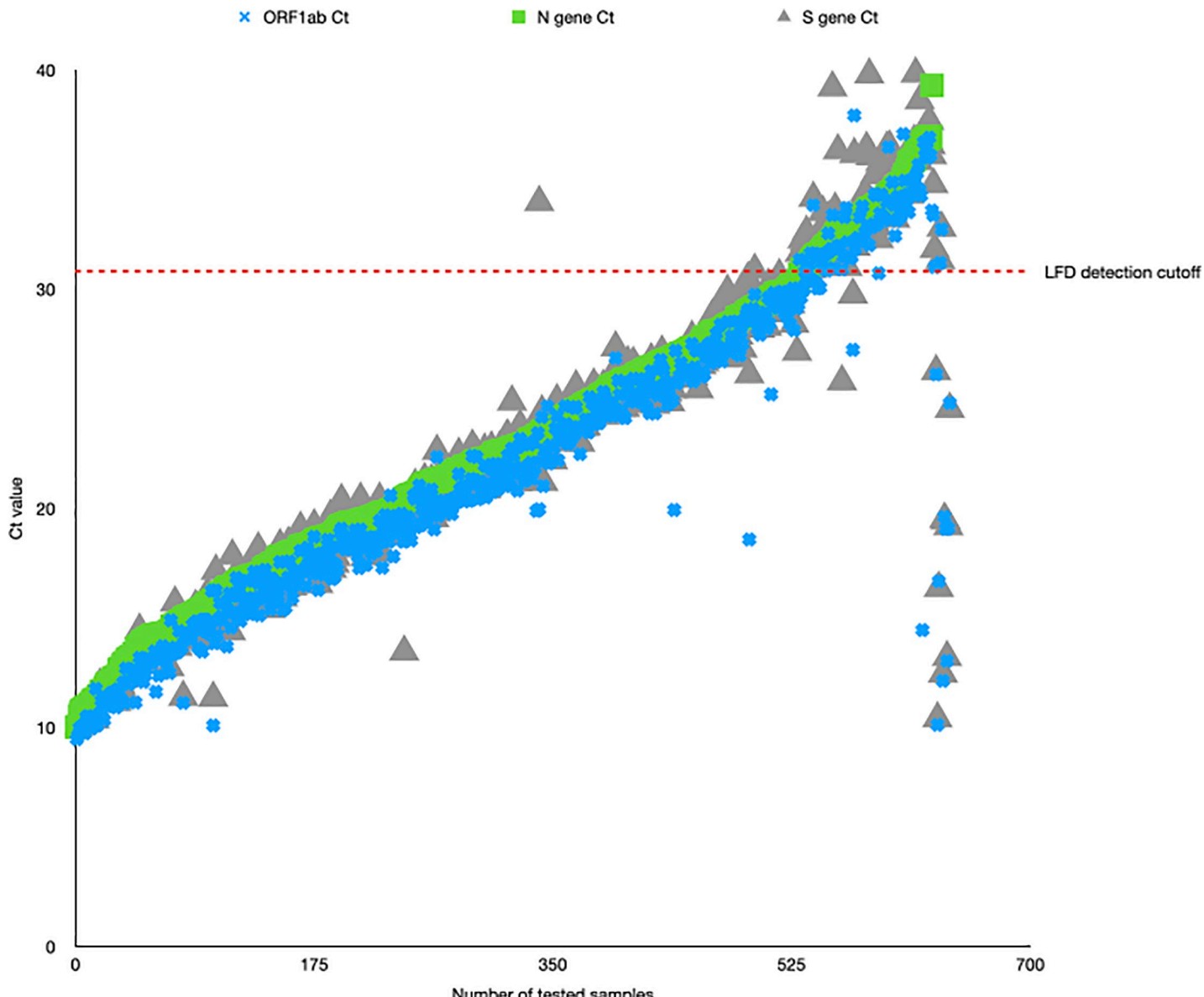

**Fig 2. Graph plotting raw Ct values (y-axis) for all 641 positive samples in the Birmingham Turnkey lab (y-axis).** Ct values for each of the targets (Orf1, N, S) are plotted, with a sample only called positive if at least 2 of the 3 targets have a Ct <35. The red line indicates the N gene Ct value equating to 100 viral copies/ml, the previously determined LoD for the Innova LFD. Ct, cycle threshold; LFD, Lateral Flow Device; LoD, Limit of Detection.

home in the lead up to the 2020 holiday period. A total of 7,183 students were tested over an 8-day period, with 2 positive tests; a prevalence of 0.03% (95% CI 0.02% to 0.10%). Approximately 10% of tests carried out across the study were sent to an onsite Pillar 2 testing lab (termed Turnkey) for RT-PCR validation. The validation, carried out using the ThermoFisher COVID-19 TaqPath assay, confirmed the 2 positive LFD results. This gave a false positive rate of rate 0%, therefore indicating a 100% (95% CI 99.48% to 100.00%) specificity of the test in this study. The high specificity rate suggests that what we are seeing are true positive samples, and this is exemplified in the recent report published by the DHSC suggesting an LFD specificity rate of 99.72% [12].

There has been some controversy over sensitivity and specificity of the Innova LFD platform [13,14]. Importantly, the studies that are often cited in such controversies relied on RT-PCR analysis of a swab taken on a separate day, whereas our study analysed the same prepared sample that provided a positive LFD result. Swabbing technique can largely impact the accuracy of diagnostic platforms; research shows that the accuracy of RT-PCR testing is significantly increased when the swab is taken by a trained scientist (79%) when compared to swabs taken by members of the public (58%) [7]. As such, a large part of the variation in sensitivity and specificity rates may be confounded by differences in swabbing technique, rather than by the technical limitations of the Innova LFD platform itself. From a public health perspective, our positive sample size was thankfully small; however, for a true validation of false positivity, an ideal dataset would include many more positive samples and, importantly, rely on a single swab to reduce variation.

By screening samples that were deemed negative by LFD, RT-PCR validation detected 6 samples that were determined to be false negatives (Table 1). This suggests a true prevalence of 0.86% (95% CI 0.40% to 1.86%) within the cohort tested and when extrapolating from these data, represents 60 potentially missed positive cases. Using this information, we explored the LOD for LFDs. Given the barriers to culturing SARS-CoV-2, we did not determine a viral titre for these samples; however, we correlated N gene Ct values with viral titre by using the Qnostics SARS-CoV-2 analytical Q-panel. This validated the limit of detection of Innova LFD devices as 100 pfu/ml, which correlated to an N gene Ct value of approximately 30. Unfortunately, the Qnostics panel contains irradiated virus which as a result cannot be detected on the Innova LFD, meaning a cross validation of the LoD could not be performed on matching material; however, our data are concordant with larger-scale studies which have comparatively reported the LoD of the Innova test [7,15,16].

Importantly, all of the false negative samples had N gene Cts above or close to 30 and therefore above or close to the limit of detection for the Innova LFDs. These may be representative of individuals at the very early or very late stages of infection, who may have a relatively lower titre of SARS-CoV-2 in the swabbed sites at the time the swab was taken. It is therefore imperative that individuals undergo regular, routine testing, to accurately remove infected individuals from community transmission pools.

First reports of variants of concern (VOCs) were published following the closure of this test site in December; however, the UK VOC (also referred to as the Kent VOC) was known to be circulating in late September [17]. Since then, numerous other VOCs have been identified through genomic sequencing of positive cases worldwide; most notably with in the UK with COVID-19 Genomics UK Consortium (COG-UK) [18]. While this study did not look at VOCs specifically, research carried out by PHE showed that the Innova LFD platform could detect the UK VOC [19]. Recent emergence of the South African VOC can also be detected on this platform [20]. With these datasets in mind, we would suggest that both the UK and South African VOCs can be reliably detected using the Innova LFD platform but as more variants emerge, these will require their own rapid evaluation by relevant public health bodies in their respective countries. The caveat to this is whether the comparative Ct values and LOD are the same for SARS-CoV-2 variants as they are with the wild type, something out of scope in this project.

Short of national restrictions, control of case numbers within a population relies heavily on diagnostic testing capacity. Multipronged approaches to mass testing using a suite of platforms including LFDs will be a huge benefit to public health and SARS-CoV-2 surveillance, provided the tests are deployed correctly and that public health messaging is clear and accurate. The Innova LFD device should "not" be considered as a green-light test—that is to say that a result from an LFD should not allow individuals to participate in an activity that they would

otherwise not participate in, if their infective status was not known. The LFD device can provide an individual with an idea of their colonisation state "at that time," but as shown by the limit of detection, individuals with a low associated N gene Ct (e.g., people at the start or end of an infection) may test negative. However, when used regularly, in the correct red-light fashion, LFDs can be a highly effective tool in reducing overall community burden, with a particular benefit to places of work and study and any other venues that would have a relatively high proportion of attendance despite potential COVID-secure restrictions.

## Conclusions

Our data show that the Innova LFD can successfully detect SARS-CoV-2 infection in people with a viral titre above approximately 100 viral copies/ml. However, as determined at our site using the ThermoFisher COVID-19 TaqPath assay, it is incapable of detecting infection at comparable PCR Ct values of 30 and over. These levels of infection are indicative of very early or very late stages of infection, and as such, we would strongly recommend that LFD testing is used to screen people at very regular frequency and that a negative result should not be used to determine that someone is free from SARS-CoV-2 infection.

## Supporting information

**S1 Table. Raw PCR Ct values for all SARS-CoV-2–positive samples analysed in the University of Birmingham Pillar 2 testing lab between October 1 and November 5, 2020.** (XLSX)

## Author Contributions

**Conceptualization:** Alex Richter, Alan McNally.

**Data curation:** Jack Ferguson, Steven Dunn, Angus Best, Jeremy Mirza, Alan McNally.

**Formal analysis:** Jack Ferguson, Steven Dunn, Angus Best, Benita Percival, Michael Kidd, Jonathan Deeks, Alan McNally.

**Funding acquisition:** Alex Richter, Alan McNally.

**Investigation:** Jack Ferguson, Steven Dunn, Angus Best, Jeremy Mirza, Benita Percival, Megan Mayhew, Oliver Megram, Fiona Ashford, Thomas White, Emma Moles-Garcia, Liam Crawford, Alan McNally.

**Methodology:** Jack Ferguson, Steven Dunn, Angus Best, Jeremy Mirza, Benita Percival, Megan Mayhew, Oliver Megram, Fiona Ashford, Thomas White, Emma Moles-Garcia, Liam Crawford, Andrew Bosworth, Michael Kidd, Alex Richter, Jonathan Deeks, Alan McNally.

**Project administration:** Jack Ferguson, Steven Dunn, Tim Plant, Andrew Bosworth, Michael Kidd, Alex Richter, Alan McNally.

**Supervision:** Jack Ferguson, Steven Dunn, Angus Best, Tim Plant, Andrew Bosworth, Michael Kidd, Alex Richter, Alan McNally.

**Writing – original draft:** Alex Richter, Jonathan Deeks, Alan McNally.

**Writing – review & editing:** Jonathan Deeks, Alan McNally.

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
