## [Editor Report · Decision Letter 0]

29 Jan 2021

Dear Dr. McNally, 

Thank you for submitting your manuscript entitled "Validation testing to determine the effectiveness of lateral flow testing for asymptomatic SARS-CoV-2 detection in low prevalence settings" for consideration as a Research Article by PLOS Biology.

Your manuscript has now been evaluated by the PLOS Biology editorial staff and I am writing to let you know that we would like to send your submission out for external peer review. Given the timeliness of the topic and also in your best interest, it would not be productive to invite a major revision, but if reviewers consider that the work almost ready to be published, we would be happy to move forward. 

Please re-submit your manuscript within two working days, i.e. by Jan 31 2021 11:59PM.

Kind regards,

Paula 

---

Associate Editor

PLOS Biology

---

## [Decision Letter · Decision Letter 1]

12 Mar 2021

Dear Dr. McNally,

Thank you very much for submitting your manuscript "Validation testing to determine the effectiveness of lateral flow testing for asymptomatic SARS-CoV-2 detection in low prevalence settings" for consideration as a Research Article by PLOS Biology. As with all papers reviewed by the journal, yours was evaluated by the PLOS Biology editors as well as by an Academic Editor with relevant expertise and by independent reviewers. The reviewers appreciated the attention to an important topic. 

Based on the reviews (below), we will probably accept this manuscript for publication, provided you satisfactorily address the points raised by the reviewers. Please also make sure to address the following data and other policy-related requests.

In particular, reviewer #1's major concern is the lack of empirical validation of the presence of infectious virus in the tested samples. We understand that this may not be possible to do retrospectively, but you should then discuss the implications and limitations of this study. For this issue, the Academic Editor also recommends to find in the literature examples of measurements of the same samples of SARS-CoV-2 for infectivity and for the levels of RNA as measured by PCR. Both reviewers also agree that you should add a discussion, add references, and provide clarifications throughout the manuscript. Reviewer #1 also says that it is critical to determine the LOD of both devices in parallel with the same panel to make sure the PCR threshold is accurately estimated, as this is crucial for the theoretical evaluation and the overall conclusions of the study. Please address the reviewers' concerns.

In order to make the manuscript more accessible to a general readership, we think that it would help adding a few sentences at the end of the abstract to explain the implications that Lateral Flow Devices do not detect infections presenting with PCR Ct values over 29-30, what % of this population was detected and undetected, and if any recommendation can be gleaned from it. We also recommend a more declarative title modifying it to include an idea of the conclusions of the work.

Please also provide a blurb which will be included in our weekly and monthly Electronic Table of Contents, sent out to readers of PLOS Biology, and may be used to promote your article in social media. The blurb should be about 30-40 words long and is subject to editorial changes. It should, without exaggeration, entice people to read your manuscript. It should not be redundant with the title and should not contain acronyms or abbreviations. 

Please also address the following data and other policy-related requests.

ETHICS STATEMENT:

-- Please include information about the form of consent (written/oral) given for research involving human participants. All research involving human participants must have been approved by the authors' Institutional Review Board (IRB) or an equivalent committee, and all clinical investigation must have been conducted according to the principles expressed in the Declaration of Helsinki.

DATA POLICY:

Regardless of the method selected, please ensure that you provide the individual numerical values that underlie the summary data displayed in the following figure panels as they are essential for readers to assess your analysis and to reproduce it: Figure 1. We note that you have provided the data, but please note that the numerical data provided should include all replicates AND the way in which the plotted mean and errors were derived (it should not present only the mean/average values).

**Please also ensure that figure legends in your manuscript include information on where the underlying data can be found**, and ensure your supplemental data file/s has a legend.

We expect to receive your revised manuscript within two weeks.

*Published Peer Review History*

*Early Version*

Sincerely,

Paula

---

Associate Editor,

pjaureguionieva@plos.org,

PLOS Biology

Reviewer remarks:

Reviewer #1: SARS-CoV-2 testing optimization. 

Reviewer #2: Diagnosis and surveillance of communicable diseases. 

Reviewer #1: Ferguson et al. compare the effectiveness of a lateral flow device with PCR for large-scale screening of asymptomatic individuals in the UK. Test results obtained with the LFD and PCR were compared, and findings were extrapolated to estimate how many false negative results would theoretically have been obtained with the LFD in comparison to PCR. This is an important study providing empirical data on performance of LFD in comparison to PCR. However, I think there is more potential in this work if the authors would have validated their findings with infectivity assay and by placing their findings in a broader context by discussing the implications and limitations of the current study. Moreover, the limit of detection experiment needs to be clarified.

Major

1. My major concern regarding this study is the lack of empirical validation of the presence of infectious virus in the tested samples. The authors conclude that the LFD could not detect infections with Ct values over Ct 29-30. Based on previous studies we know that this is in line with the typical threshold of samples containing infectious virus, but it would have been much stronger if the authors could provide this evidence. Ct values can differ drastically between labs and different PCR platforms, and therefore it would be preferable to empirically determine the correlation between infectivity and Ct values for this particular study. I understand that these experiments are very challenging and that they might not be possible retrospectively, but at the minimum the authors should discuss the true implications and limitations of this study, which is currently lacking in the manuscript. Particularly, a discussion on the application of LFD for SARS-CoV-2 screening is needed as it can still be used as an effective tool despite the lower sensitivity in comparison to PCR. Please cite previous studies accordingly throughout the manuscript.

2. The LOD experiments require some clarifications. The authors tested the Qnostics Q-panel to estimate the Ct value that would match the LOD of the LFD (100 copies/mL). However, it seems striking to me that not all 3 PCR targets were able to consistently detect the panel at 100 copies/mL, while the LOD for the PCR should be much more sensitive than the LFD. It looks like the panel consists of inactivated virus, but I wonder if the reported concentrations are for PFU/mL or RNA copies/mL. This would make a huge difference as the authors report in the introduction that the LOD of the LFD is 100 PFU/mL or 100.000 RNA copies/mL. It would be critical to determine the LOD of both devices in parallel with the same panel to make sure the PCR threshold is accurately estimated, as this is crucial for the theoretical evaluation and the overall conclusions of the study. Please also clarify whether the LOD was tested in 3 (main text) or 5 (legend table 3) replicates.

3. I would suggest to extend the introduction by (1) describing background on the use and performance of lateral flow tests for SARS-CoV-2 screening of asymptomatic individuals (key publications are missing), (2) stating the overall aims of the study more clearly at the end, (3) summarizing the main findings.

4. The methods currently lack (1) an ethics statement, and (2) brief description to the Innova, RNA extraction, and TaqPath protocols.

Minor

1. Title suggestion: Validation of the effectiveness of the Innova lateral flow device for asymptomatic SARS-CoV-2 detection in low prevalence settings. Perhaps even making the title more specific by replacing "effectiveness" by "sensitivity" or another more specific term.

2. Line 91: Please specify how much the sample was diluted by adding 500 uL of water. Was this dilution factor taken into account when comparing LFD and PCR results?

3. Line 173: please rephrase

4. Fig1: remove zeros after decimal on Y-axis. X-axis label should read "number of tested samples"

5. Line 35 and Line 190: Ct values can differ significantly between different labs and PCR platforms. Please specify that the observed Ct value threshold is specific for the used PCR platform.

Reviewer #2: In this article, Ferguson et al. provide useful information on the clinical performance of a widely used LFD in the UK, the Innova LFD. The article was clear and well-written, but would benefit from additional methodological detail, and a Discussion section. Comments are below:

Introduction

1. The first sentence needs a reference.

2. Can the authors provide some more context about why the Innova LFD was 'principal' in being rolled out? This isn't immediately obvious for a non-UK audience.

3. 'Data from the city council shows that 71,684 LFD tests were performed alongside 51,855 gold-standard PCR tests' - this makes it sounds like tests were performed at the same time in the same individuals - can the authors clarify the number of individuals tested. Not clear from this sentence.

4. Lines 49-50 - needs ref.

5. Line 55 - what is the FALCON trial?

6. Lines 56-57 needs references.

7. Lines 60-61 - why can't the authors access this information - bit speculative otherwise.

Methods

8. Lines 79-80 - what was the result recording app? More detail required here.

9. Please provide information on what 'pillar' laboratories are - again, not clear to a non-UK audience.

10. Line 86: Why were 90 samples selected - was this figure derived statistically?

11. Line 86: How do the authors know that the solution is saline? Please reference the Innova IFU (I'm aware that other LFD solutions are not saline)

12. Line 99 - should read 720/7187

Results

13. Line 119: please provide criteria for deeming a sample void.

14. Line 144: RT-PR should be RT-PCR

15. Line 159: 'From our data the Lateral Flow Device test yield is 2.8 per 10,000 (0.3 to 10 per 10,000).' - not clear what the units here are? Tests?

Conclusions

16 Line 190: 'However it is incapable of detecting infection at comparable PCR Ct values of 30 and over.' - this needs qualifying to say that this is only based on an analysis using the TaqPath assay - it may be different with other assays.

I was a bit surprised not to see a Discussion here - there are obviously lots of facets to discuss (implications of false pos/neg; logistical roll out of tests; impact of variants of concern on LFD detection; frequency of testing). The authors should include a broader discussion of their findings here - not enough just to have a brief concussion section.

---

## [Editor Report · Decision Letter 2]

1 Apr 2021

Dear Dr. McNally,

On behalf of my colleagues and the Academic Editor, Bill Sugden, I am pleased to say that we can in principle offer to publish your Research Article "Validation testing to determine the sensitivity of lateral flow testing for asymptomatic SARS-CoV-2 detection in low prevalence settings: testing frequency and public health messaging is key." in PLOS Biology, provided you address any remaining formatting and reporting issues. Please ensure that the figure legends in your manuscript include information on where the underlying data can be found. 

The rest of the formatting and reporting issues will be detailed in an email that will follow this letter and that you will usually receive within 2-3 business days, during which time no action is required from you. Please note that we will not be able to formally accept your manuscript and schedule it for publication until you have made the required changes.

PRESS

Thank you again for supporting Open Access publishing. We look forward to publishing your paper in PLOS Biology. 

Sincerely, 

Paula

---

Paula Jauregui, PhD 

Associate Editor 

PLOS Biology